# Competition between crystal growth and intracrystalline chain diffusion determines the lamellar thickness in semicrystalline polymers

Martha Schulz [1], Mareen Schäfer[1], Kay Saalwächter [1✉] & Thomas Thurn-Albrecht [1✉]

The non-equilibrium thickness of lamellar crystals in semicrystalline polymers varies significantly between different polymer systems and depends on the crystallization temperature $T_c$. There is currently no consensus on the mechanism of thickness selection. Previous work has highlighted the decisive role of intracrystalline chain diffusion (ICD) in special cases, but a systematic dependence of lamellar thickness on relevant timescales such as that of ICD and stem attachment has not yet been established. Studying the morphology by small-angle X-ray scattering and the two timescales by NMR methods and polarization microscopy respectively, we here present data on poly(oxymethylene), a case with relatively slow ICD. It fills the gap between previously studied cases of absent and fast ICD, enabling us to establish a quantitative dependence of lamellar thickness on the competition between the noted timescales.

[1] Institut für Physik, Martin-Luther-Universität Halle-Wittenberg, 06099 Halle, Germany. ✉email: kay.saalwaechter@physik.uni-halle.de; thomas.thurn-albrecht@physik.uni-halle.de

The characteristic morphological feature of semicrystalline polymers crystallized from the melt is a nanoscopic two-phase structure of thin lamellar crystals separated by disordered amorphous layers, which contain the entanglements retained during crystallization. This morphology is largely responsible for the favorable mechanical properties of semicrystalline polymers[1]. It has been a classical question in polymer physics which factors control the thickness of the crystalline layers, resulting in a number of crystallization models without reaching final consensus[2,3].

Most crystallization models start from the assumption that the semicrystalline morphology is a non-equilibrium structure, which is experimentally supported by the observation of a melting point depression that depends on thermal history, specifically the crystallization conditions. Structurally the melting point depression is explained by a finite crystal thickness[1]. As a consequence, for a given crystallization temperature $T_c$ there is a minimum stable crystal thickness. To explain the selection of a relatively well-defined crystal thickness during crystallization of chemically uniform linear polymers, a second criterion defining an upper limit for the thickness is required. At this point, the assumptions made by different models diverge. The classical approach assumes that the crystal thickness is kinetically selected. The crystals with the thickness that grow the fastest, dominate[4–8], and once a stable crystal has formed, it is assumed that no further structural changes will take place. Multistage models on the other hand assume that crystal growth happens in several stages and is coupled to crystal reorganization processes. Different mechanisms have been suggested—without reaching a final agreement—to limit reorganization to a certain thickness, such as thickness-dependent stability of different crystal phases[9] or mesophases[10,11] or thickness-dependent intracrystalline chain diffusion (ICD)[12–14]. All these models primarily aim at an explanation of the temperature dependence of the crystal thickness of a given semicrystalline polymer. They disregard to the most part variations of crystal thickness between different polymers as well as the question of what determines the thickness of the amorphous layers and therefore the overall crystallinity.

In view of this incomplete understanding, we started a series of investigations with the aim of providing a broader perspective on the formation of the semicrystalline morphology by comparing polymers with and without ICD. Our starting point was an old observation by Boyd[15,16] that relates the crystallinity of a polymer to the existence of a so-called $\alpha_c$-relaxation process. These relaxation processes are a unique feature of polymer crystals and originate from conformational defects moving through the crystals. They enable ICD, as shown later directly by advanced nuclear magnetic resonance (NMR) methods[17]. Generally, polymers with ICD (crystal-mobile) show a higher crystallinity (> 50%) than polymers without ICD (crystal-fixed). For the specific case of poly(1-butene), an important contribution of the ICD to the crystal thickness was suggested. This polymer shows two crystal structures, of which one is crystal-fixed while the other one is crystal-mobile[18,19]. The relevant observation was that direct crystallization into the crystal-fixed form I, either by crystallization from solution or by choosing a sample with tacticity defects, led to much thinner crystals than the usual pathway, in which crystallization proceeds via the crystal-mobile form II, followed by a solid–solid transition into form I[19,20]. For the latter case, the crystal thickness also showed a stronger dependence on the crystallization temperature, presumably caused by the stronger effect of ICD at high temperatures. However, the question of what finally limits the crystal thickness was not specifically addressed.

Previously, we systematically compared a pair of crystal-fixed and crystal-mobile model polymers making use of new experimental developments in SAXS, NMR, and differential scanning calorimetry (DSC). Our experiments led us to the hypothesis that generally, the morphology of semicrystalline polymers results from the interplay or competition between the kinetics of crystal growth and ICD leading to different morphological characteristics of crystal-fixed and crystal-mobile polymers[21]. The crystallization of a crystal-fixed polymer like poly($\epsilon$-caprolacone) (PCL) results in the formation of marginally stable crystallites of well-defined thickness, which reorganize constantly during heating. We could, later on, confirm this result by fast scanning calorimetry[22]. A crystal-mobile polymer like poly(ethylene oxide)(PEO) on the other hand shows a well-defined thickness of the amorphous regions and crystalline lamellae that are stable over a wide temperature range. Detailed analysis of NMR data reflecting the timescale of ICD in the temperature range of crystallization showed that indeed for PEO the ICD is so fast that it can cause reorganization over a very small nanometre-sized reorganization zone directly behind the growth front and practically simultaneously with crystal growth[21]. From these results, we concluded that in crystal-mobile polymers the morphology is controlled by a minimum value of the amorphous thickness related to the entanglement density in the amorphous regions.

In order to enable a more quantitative description of the above-mentioned competition between crystal growth and ICD we introduced three parameters describing the typical timescales. As depicted in Fig. 1a, we describe the timescale of crystallization by the layer crystallization time $\tau_{lc}$, the time during which the crystal grows on average by one molecular layer. $\langle\tau_c\rangle$ and $\tau_{stem}$ on the other hand are the characteristic timescales of the defect dynamics underlying the ICD. Here, $\langle\tau_c\rangle$ is the so-called jump correlation time as probed by NMR, i.e., the average time between two helical defect jumps, which corresponds to the $\alpha_c$-relaxation

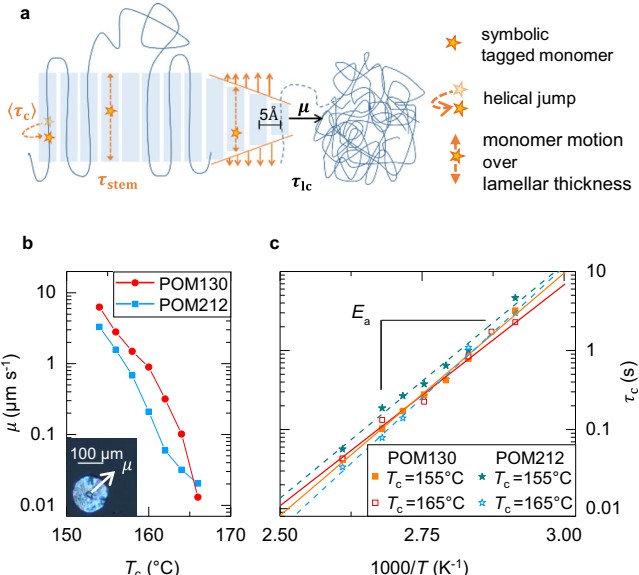

**Fig. 1 Competing timescales of crystal growth ($\tau_{lc}$) and intracrystalline chain dynamics ($\langle\tau_c\rangle$, $\tau_{stem}$). a** Schematic illustration of the crystallization process with $\tau_{lc}$, $\langle\tau_c\rangle$, and $\tau_{stem}$. **b** Growth velocity $\mu$ of poly(oxymethylene) for two molecular weights (POM130 and POM212) as a function of crystallization temperature $T_c$. The inset illustrates how $\mu$ is determined from the growth of spherulites during isothermal crystallization. **c** Arrhenius temperature dependence of the jump correlation time $\langle\tau_c\rangle$ for two molecular weights (POM130 and POM212) isothermally crystallized at different $T_c$.

**Table 1 Sample characteristics.** The molecular weights and therefore also $R_{max}$ are nominal values based on the GPC calibrations given in the text. Only for PCL a correction factor of 0.56 is known, which would reduce $R_{max}$ to a value of 413 nm[75]. The melting temperatures (peak maxima) and crystallinities were obtained by DSC heating scans. The samples were cooled from the melt and heated at a rate of 10 K min$^{-1}$.

| Sample | Supplier/industry name | $M_w$ (kg mol$^{-1}$) | $M_n$ (kg mol$^{-1}$) | $R_{max}$ (nm) | $M_e$ (kg mol$^{-1}$) | $T_m$ (°C) | $X_c$ (%) |
|---|---|---|---|---|---|---|---|
| POM130 | DuPont/Delrin® 500P NC010 | 130 | 39.9 | 255 | 2.64[76] | 177 | 61 |
| POM212 | DuPont/Delrin® 100P NC010 | 212 | 62.8 | 402 | | 179 | 59 |
| PCL138 | Sc. Polym. Products, Inc./- | 138 | 97.4 | 738 | 2.5 ± 0.5[41] | 58 | 40 |
| PEO180 | PSS Polymer Standards Service/- | 180 | 148 | 936 | 2.00[76] | 69 | 74 |

time. In contrast, $\tau_{stem}$ represents the time, during which a given tagged monomer moves in a diffusive fashion over a distance equal to the crystal thickness $d_c$ by successive helical jumps, which are in turn mediated by defects traveling quickly over the length of the stem. In this way, the crystalline stem is renewed. The prediction of $\tau_{stem}$ requires the use of a specific model, e.g., 1D diffusion, possibly constrained by loop size and entanglements in the amorphous phase[23]. Our previous experiments on PCL and PEO correspond to the cases of non-existing (or very slow) and very fast ICD, i.e., $\langle \tau_c \rangle \gg \tau_{lc}$ and $\langle \tau_c \rangle \ll \tau_{lc}$, respectively. $\langle \tau_c \rangle$ is measured on the fully crystallized sample. As we cannot exclude that the ICD is faster directly behind the growth front, the measured $\langle \tau_c \rangle$ is an upper estimate for the relevant parameter, but this does not harm the arguments in general.

Here, we present a set of experiments designed as a critical test of the hypothesized competition between crystal growth and ICD by extending our previous studies to a polymer with ICD on an intermediate timescale, namely poly(oxymethylene) (POM). This choice of sample enables us to establish a quantitative dependence of lamellar thickness on the competition between the noted timescales. In such a case we expect intermediate crystal thicknesses, and additionally, the opposite temperature dependencies of $\langle \tau_c \rangle(T)$ and $\tau_{lc}(T)$ should play an important role. Furthermore, we extend our previous static SAXS experiments to time-dependent measurements using a position-dependent detector, which enables us to observe the thickening of lamellar crystals directly during different stages of crystallization. The timescales of crystal growth and ICD are characterized by optical microscopy and solid-state NMR, respectively.

## Results

**Characteristic timescales.** We start with the determination of the characteristic times in POM. Following ref. [21] the layer crystallization time $\tau_{lc}$, during which a crystal grows on average over a distance corresponding to one molecular layer, can be calculated from the crystal growth velocity $\mu$,

$$\tau_{lc} = \frac{5 \text{ Å}}{\mu} \quad (1)$$

assuming a typical intermolecular distance of the order of 5 Å. $\mu$ was measured by optical microscopy. Figure 1b shows $\mu$ as a function of $T_c$ for two molecular weights, POM130 and POM212 (cf. Table 1). Corresponding data for PCL and PEO were already published and can be found in Supplementary Table 2.

Previous investigations have shown that POM belongs to the class of crystal-mobile polymers and that its ICD is much slower than in PEO[17]. Most of the corresponding experiments were performed in the 1960s by mechanical and dielectric measurements, resulting in a wide range of reported activation energies from $E_a = 88$ to $328$ kJ mol$^{-1}$ [24–30]. The first NMR-based value was reported by Kentgens using 2D exchange measurements and the copolymer Hostaform as a sample, $E_a = 83 \pm 68$ kJ mol$^{-1}$ [31]. Higher precision was achieved later by Schmidt–Rohr and Spiess

with a value of $E_a = 83 \pm 8$ kJ mol$^{-1}$ for a not further specified POM homopolymer[32].

To provide a detailed and reliable characterization of the ICD for the same samples as used for the structural analysis, we performed NMR experiments on isothermally crystallized POM samples, using the $^{13}$C MAS CODEX technique[33], which probes slow segmental reorientations. The analysis of NMR spectra measured at different temperatures (here 70–110 °C) allows the determination of $\langle \tau_c \rangle(T)$ and of the activation energy describing its temperature dependence

$$\langle \tau_c \rangle = \tau_0 \cdot \exp \frac{E_a}{RT} \quad (2)$$

$E_a$ and $\tau_0$ were determined for different $T_c$s and both molecular weights. Exemplary results are shown in Fig. 1c, the full set of resulting values are listed in Supplementary Table 1. As the samples with different $T_c$ have different crystal thickness $d_c$ (see below) this analysis also reveals if $\langle \tau_c \rangle$ depends on $d_c$. While the activation energies vary by about 10% from sample to sample, we could not observe a systematic dependence of $\langle \tau_c \rangle$ on $d_c$, different from the case of PEO[34]. Our data do not allow for conclusions on a potential molecular weight effect. The average values are $E_a = 113$ kJ mol$^{-1}$ and $\tau_0 = 2.0 \times 10^{-17}$ s for POM130 and $E_a = 117$ kJ mol$^{-1}$ and $\tau_0 = 5.6 \times 10^{-18}$ s for POM212. For consistency, we cross-checked the results by dynamic mechanical measurements and found similar results. Details are given in the Supplementary Information.

The typical time scale of crystal reorganization can be estimated from the correlation time $\tau_c$ measured by NMR. $\langle \tau_c \rangle$ corresponds to the average residence time of a monomer (and thus of the chain) in a given helical raster[34]. For an $n_m$-helix with $n$ monomers per $m$ turns over a lattice distance $c$, the corresponding monomer jump distance is $\Delta z_c = c/n$ ($\Delta z_c = 0.279$ nm for the $7_2$ helix in PEO[35] and $\Delta z_c = 0.192$ nm for the $9_5$ helix in POM[36]. The value given for PEO in ref. [21] contained an erroneous factor of 3.5.) We estimate the time $\tau_{stem}$ within which a monomer as part of the crystal stem diffuses over a distance equal to the crystal thickness by successive helix jumps as

$$\tau_{stem} \approx \langle \tau_c \rangle \cdot d_c^2 / \Delta z_c^2. \quad (3)$$

Here, we assumed a random walk of $N = \tau_{stem}/\langle \tau_c \rangle$ steps of size $\Delta z_c$. The squared distance traveled is $d_c^2$. For $d_c$ we use the values of the lamellar thickness obtained by SAXS measurements after isothermal crystallization as shown below. The corresponding data are listed in Supplementary Table 3. Eq. (3) is an approximation for the early stage of growth, during which constraints by neighboring lamellar crystals are still weak.

Together with previously published data for PEO, with fast ICD, and PCL, for which we could exclude any dynamics up to a timescale of 1 s, we can now compare the timescales for ICD (range between $\langle \tau_c \rangle$ and $\tau_{stem}$) and crystal growth ($\tau_{lc}$) in the temperature range of crystallization for all three polymers[21,34,37]. The result is shown in Fig. 2. For PCL we used the NMR

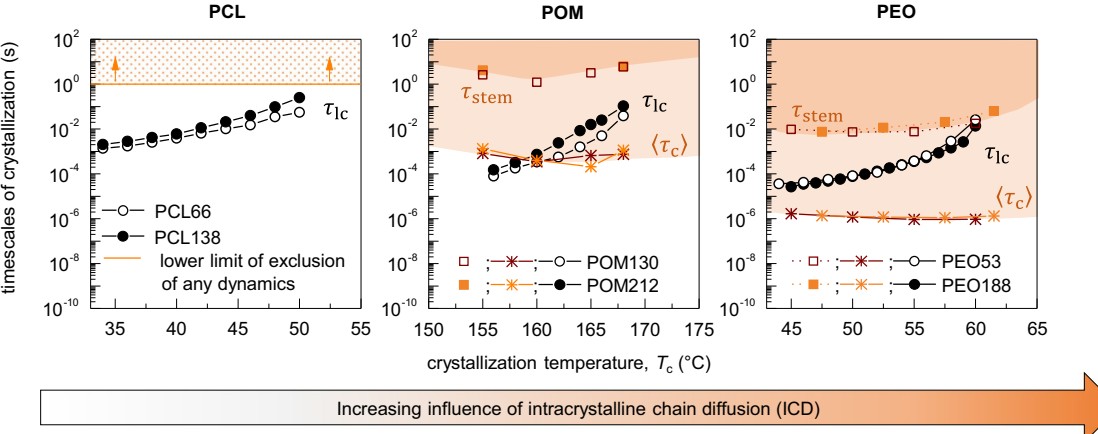

**Fig. 2 Comparison of the timescales underlying intracrystalline chain diffusion and crystal growth for PCL, POM, and PEO.** $\tau_{lc}$ corresponds to the time during which the crystal grows by one molecular layer according to Eq. (1). $\langle\tau_c\rangle$ is the average residence time between two helical jumps calculated with Eq. (2) and the values given in the Supplementary Table 3. $\tau_{stem}$ denotes the time during which a chain in the crystal diffuses over a distance equal to the lamellar thickness $d_c$, estimated by Eq. (3). $\langle\tau_c\rangle$ can be considered as a lower and $\tau_{stem}$ as an upper limit of the timescale of crystal reorganization enabled by the $\alpha_c$-relaxation. For PCL the solid line shows the NMR detection limit for $\langle\tau_c\rangle$ due to a possible, undetectably slow $\alpha_c$-relaxation and thus ICD[37]. Figure partially adapted from https://pubs.acs.org/doi/10.1021/acs.macromol.8b01102, further permission related to the material excerpted should be directed to the ACS.

detection limit for ICD as a lower limit for $\langle\tau_c\rangle$. Clearly, it is much larger than $\tau_{lc}$ and the timescales of crystal growth and any possible reorganization are well separated. In contrast, for POM and PEO the timescales of crystallization given by $\tau_{lc}$ and the timescale of reorganization (given by the band between $\langle\tau_c\rangle$ and $\tau_{stem}$) overlap. However, while for POM $\langle\tau_c\rangle$ becomes smaller than $\tau_{lc}$ only for the higher crystallization temperatures above 160 °C, for PEO $\tau_{lc}$ lies well above $\langle\tau_c\rangle$ for the whole temperature range and even becomes comparable to $\tau_{stem}$ for the higher crystallization temperatures. Consequently, we confirm that POM fills the relevant gap between PCL and PEO with regard to the ratio between $\tau_{lc}$ and $\langle\tau_c\rangle$, and enables us to establish quantitatively the role of the ICD for crystallization and structure formation. Following the direction of the arrow in Fig. 2 from PCL over POM to PEO, we expect an increasing effect of the ICD on crystal growth and the semicrystalline morphology.

**Semicrystalline morphology.** To investigate the effects of ICD on the morphology we performed SAXS measurements during and after isothermal crystallization at different $T_c$. Based on a recently refined SAXS analysis[38,39] we obtain the Porod parameter $P$ as a measure of the specific inner surface between crystalline and amorphous regions, the average thicknesses $d_{c/a}$ of the crystalline and amorphous regions together with their distribution widths $\sigma_{c/a}$ in the lamellar stack and the long period $L$.

Figure 3 shows the results. For each sample system measurements for two $T_c$s are exemplarily shown, corresponding to the lower (supercooling $\Delta T \approx 20$ K) and the upper limit ($\Delta T \approx 5$ K) of the experimentally accessible range of isothermal crystallization. The measurements are arranged in such a way that from top to bottom, following the arrow on the left-hand side of Fig. 3, we expect a growing influence of the ICD. As the Porod parameter $P$ is proportional to the amount of crystalline-amorphous interface per volume, we can follow the crystallization process and identify the end of the primary crystallization, which is marked by a vertical, dotted line. For these time-dependent measurements, which go beyond our previous study, we used a PEO sample from a new batch with a slightly lower molecular weight $M_w$ than in Fig. 2. The PCL and POM samples are from the same batches. The sharp steps and oscillations of $P$ during a measurement series are artifacts

caused by slight changes in the alignment of the X-ray optics due to residual temperature variations in the system caused by intermediate closure times of the X-ray shutter and 24 h temperature oscillations over the course of a day.

PCL shows the already known typical structure of a crystal-fixed polymer with a linear crystallinity around 50%, a well-defined $d_c$ (small $\sigma_c$), and a broader distribution for $d_a$. $d_a$ increases only slightly with increasing crystallization temperature. As a new result, observable by the long time series in these measurements, we observe a very small increase in $d_c$ and a corresponding decrease in $d_a$.

In comparison, POM shows strong structural changes with time for both $T_c$s. For the lower crystallization temperature, these changes mostly take place after the primary crystallization. We observe not only an increase in $d_c$ and a decrease in $d_a$, but also a decreasing distribution width for $d_a$, such that the relative width $\sigma_a/d_a$ remains approximately constant. With time the morphology develops the typical morphology of a crystal-mobile polymer as observed before in PEO[21,39] with a well-defined $d_a$ (small $\sigma_a$) and a more broadly distributed $d_c$. Our observations are in keeping with previously observed long-time lamellar thickening in POM[40], for which we now clearly establish the relatively slow ICD as its origin. The comparison with the higher $T_c$ shows that $d_c$ depends much more strongly on $T_c$ than in the case of PCL. Crystal thickening takes place to a large part during primary crystallization and slows down afterwards indicating the interplay with crystal growth. Correspondingly, already during primary crystallization a crystal-mobile morphology forms. Generally, the linear crystallinity $X_c$ is higher than in the case of PCL. For $T_c = 155$ °C $X_c$ increases from 65% at the end of the primary crystallization to 77% at the last measurement point and from 74 to 81% for $T_c = 168$ °C.

In PEO we observe the typical crystal-mobile structure for both $T_c$ already during primary crystallization. In contrast to POM, a strong increase/decrease of $d_{c/a}$ can take place already during primary crystallization. Afterward, the changes slow down, and also the distributions widths $\sigma_{c/a}$ show no further significant changes. The lamellar thickness $d_c$ depends even more strongly on $T_c$ than for POM. After finishing the primary crystallization, $X_c$ changes from 65% to 74% for $T_c = 45$ °C and from 78 to 80% for $T_c = 60$ °C.

The observations for PCL and PEO confirm and extend our previous results[21]. They correspond to the limiting cases of no or

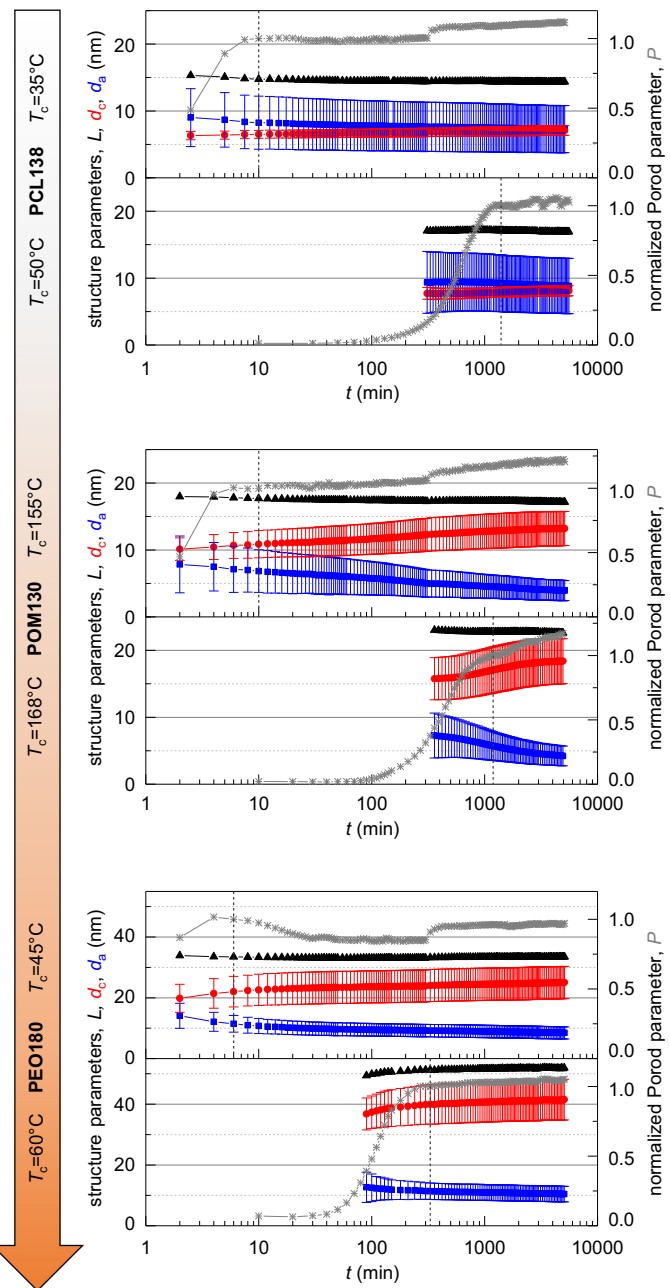

**Fig. 3 Structural parameters as obtained by SAXS during isothermal crystallization from the melt as a function of the crystallization time.** For each sample system the isothermal crystallization was performed for a high (~20 K) and a low (~5 K) supercooling $\Delta T$. The arrow represents the expected influence of the ICD according to Fig. 2. RHS $y$-axis: time-dependent Porod parameter $P$ (grey) normalized to 1 at the end of the primary crystallization process (dotted line). LHS $y$-axis: time-dependent long period $L$ (black), amorphous thickness $d_a$ (blue), and crystalline thickness $d_c$ (red). $\sigma_{c/a}$ are shown as "error bars". For PEO the scale of the left $y$-axis is increased roughly by factor 2. Source data are provided as a Source Data file.

very slow and very fast ICD, respectively, with the corresponding characteristic morphologies. The results for POM now clearly establish the timescale of ICD and its competition with the one of crystal growth as the relevant effect. We directly observe strong lamellar thickening, which for the higher $T_c$ takes place mostly during primary crystallization and for the lower $T_c$ after primary crystallization according to the relative values of the characteristic times shown in Fig. 2 and represented by the arrow in Fig. 3. Important is the observation that crystal thickening slows down around the end of the primary crystallization, which indicates that a certain limiting structure is reached, which is characterized by a well-defined minimum value of the amorphous regions, as we concluded already from our previous measurements on PEO.

We tentatively explained this limiting structure by a critical entanglement density in the amorphous phase[21,41]. Here the long-time experiments give interesting new information. The fact that lamellar thickening continues over the whole experimental time scale, shows that this limiting structure is still a non-equilibrium structure and slowly develops further if enabled by ICD. But obviously, this process is considerably hindered in a fully developed semicrystalline morphology. On the other hand, the observation of longtime lamellar thickening in PCL indicates that even in this, at first sight, crystal-fixed, polymer a very slow ICD undetectable by NMR might exist. A further comment concerns the slight decrease of the long period for most of the experiments, which had been observed before for POM and was

taken as an indication for insertion crystallization[42]. Although we cannot completely exclude the existence of such an additional process, we consider it as negligible, as it would be inconsistent with the observed increase of $d_c(t)$ and the decrease of $\sigma_a(t)$.

The most important new aspects in our data are the quantitative determination of $\langle\tau_c\rangle$ in the relevant temperature range and its evaluation in terms of the kinetics of crystal growth i.e. $\tau_{lc}$, together with the quantitative analysis of the full set of structural parameters. This much broader set of experimental data renders a comparison of the semicrystalline morphology across different polymer systems as in Fig. 3 meaningful, whereas typically in the literature the lamellar thickness $d_c$ of semicrystalline polymers is discussed only for individual polymer systems in relation to the crystallization temperature $T_c$.

A corresponding comparison of the time-dependent value of $d_c$ for all $T_c$s and all three polymer systems is shown in Fig. 4a. Additionally to the data already shown in Fig. 3, data from further $T_c$s and a second sample (POM212) are included. We observe a systematic increase of $d_c$ following the order induced by the ICD indicated as above by the orange arrow. The data suggest that the value of the crystal thickness is affected from the very beginning of the crystallization by the ICD to a degree that depends on the crystallization temperature as well as on the inherent polymer-dependent timescale of the ICD. The idea is illustrated in the inset of Fig. 4a, which also explains the different curvatures observed for $d_c(t)$ for PCL, POM, and PEO. In this picture, the crystallization process starts with an initial crystalline thickness $d_{c0}$ which then increases due to reorganization enabled by ICD but is finally restricted by the presence of neighboring crystallites and the fact that the minimum thickness of the amorphous regions is reached. Accordingly, crystal reorganization competes with crystal growth, if both take place on a similar timescale. Following this idea, Fig. 4b shows the largely different values of $d_c$ across the three different polymer systems as a function of the ratio of the characteristic times $\langle\tau_c\rangle/\tau_{lc}$, which depends on the polymer system as well as on the crystallization temperature. In addition, $d_c$ is normalized by the height of a monomer unit in direction of the stem, as this is the distance over which all monomers of a chain move during one helical jump[17], neglecting a possible chain tilt. The factors $h_{mon}$ were calculated from the length of the unit cell in the $c$-direction (PEO: $\frac{1.95}{7}$ nm[35]; POM: $\frac{1.73}{9}$ nm[36]; PCL: $\frac{1.73}{2}$ nm[43]). Reorganization on the timescale of the primary crystallization ($\tau_{lc} \gg \langle\tau_c\rangle$), as for PEO, leads to high $d_c$ values, whereas reorganization, which is slower in comparison to crystal growth as for POM ($\tau_{lc} \approx \langle\tau_c\rangle$), leads to smaller values and is eventually more restricted by neighboring crystallites. Accordingly, there is a more or less smooth progression of the $d_c$-values from POM to PEO. This scaling is only observed if $\tau_{lc}/\langle\tau_c\rangle$ is taken as the variable, not for $1/\langle\tau_c\rangle$ alone, cf. Supplementary Information Fig 5. On the other hand, the values for PCL fall out of trend in line with the fact that PCL has no or very slow ICD ($\tau_{lc} \ll \langle\tau_c\rangle$). In this case, reorganization plays no role during primary crystallization and there is only a small amount of reorganization or thickening later on. Combining SAXS with ultrafast scanning calorimetry we recently showed that in this latter case of PCL the crystalline lamellae are only marginally stable, i.e. they melt immediately upon heating and $T_m \approx T_c$[21,22]. In this case, the crystals grow obviously with very small supercooling, and the initial crystal thickness is basically controlled by thermodynamics. The corresponding effect of temperature is weak, as the measurements are shown here for $T_c = 35\,°C$ and $T_c = 50\,°C$ illustrate.

Thickening should always lead to further thermodynamic stabilization and an increased melting temperature. While we observed such effects previously in PEO[21], for POM they can be demonstrated directly as we follow the thickening process at the higher crystallization temperature. Figure 5 shows the Porod

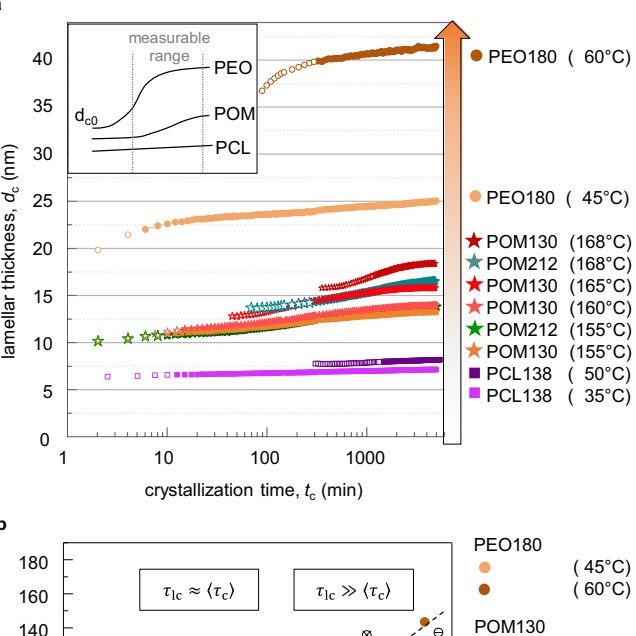

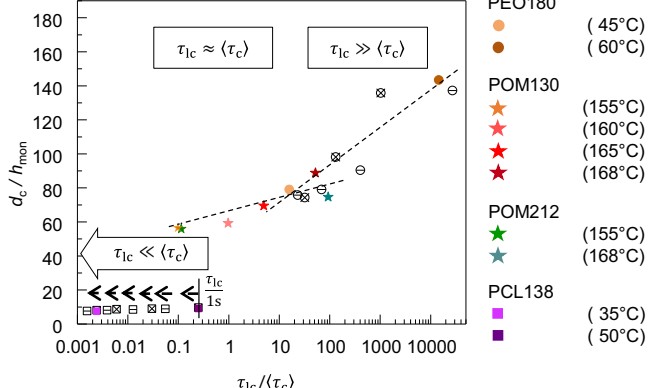

**Fig. 4 Direct comparison of the crystalline thickness $d_c$ for PCL, POM, and PEO. a** $d_c$ during isothermal crystallization for all crystallization temperatures. The orange arrow represents the increasing influence of the ICD according to Fig. 2. The change from open to closed symbols indicates the completion of the primary crystallization. **b** Normalized lamellar thickness $d_c$ at the end of primary crystallization vs. the ratio of the timescales $\tau_{lc}/\langle\tau_c\rangle$. Depending on the ratio $\tau_{lc}/\langle\tau_c\rangle$ different regimes are visible. The dotted lines are guides to the eye. The closed symbols are the data from **a**, open symbols represent already published data of PEO (open circles) and PCL (open squares) for two molecular weights (cross and line) measured at different $T_c$s[21]. For PCL a constant value of 1 s was taken for $\langle\tau_c\rangle$, corresponding to the lower limit of a possible $\alpha_c$-relaxation time and an upper limit of $\tau_{lc}/\langle\tau_c\rangle$ as indicated by the broken arrow. Source data of **a** are provided as a Source Data file.

parameter $P$ and lamellar thickness $d_c$ as resulting from in-situ SAXS experiments during stepwise heating after isothermal crystallization at $T_c = 155\,°C$ for different crystallization times $t_c$. Generally, the melting process goes along with a strong decrease of $P$ and an increase of the average value $d_c$, due to the melting of thinner lamellae. The dominant processes affecting $d_c$ during heating before final melting are indicated in Fig. 5. Indeed, with increasing crystallization time $t_c$ the melting process shifts to higher temperatures. For crystallization times of 0.5 and 5.2 h, the lamellar thickening process continues also during heating. Only for the longest crystallization time of 85 h (5100 min) the trend is inverted. Now heating leads to a decrease of $d_c$ in the temperature range below final melting, an effect well-known for PE[44] and PEO[21], which is called surface melting and caused by a local equilibrium

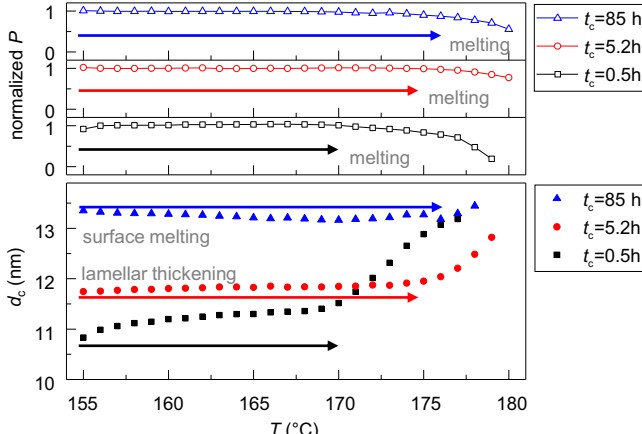

**Fig. 5 Effect of crystallization time on melting temperature.** Porod parameter $P$ and crystal thickness $d_c$ of POM130 during stepwise heating after isothermal crystallization at 155 °C for different crystallization times: $t_c = 0.5$ h (black), $t_c = 5.2$ h (red), and $t_c = 85$ h (blue). The arrows indicate the temperature range between the crystallization temperature and the temperature at which melting sets in as indicated by the decreasing values of $P$ for the different values of $t_c$. Source data are provided as a Source Data file.

between the crystals and the adjacent amorphous phase. These observations are in keeping with the hypothesis that crystal thickening is restricted by the constraints in the amorphous phase.

## Discussion
How do these observations compare with previous experimental results? Lamellar thickening, as a fundamental process occurring in semicrystalline polymers, is well-known[45]. However, direct observations including the regime of primary crystallization have been scarce[46]. Common are either DSC observations, where an increase of the melting temperature is interpreted as an indication of lamellar thickening, or SAXS measurements of the long period[47,48]. Long-time annealing experiments gave evidence for the logarithmic time dependence also observed here at times beyond the primary crystallization[49]. Nearly all experiments deal with polyethylene, a common crystal-mobile polymer, which however has the disadvantage that it is difficult to measure $\langle \tau_c \rangle$ by NMR and also that samples with well-defined molecular weight and low polydispersity are difficult to synthesize. A long-standing discussion concerned the question if on a microscopic level chain movement is caused by a sliding motion of the whole chain or induced local defects[50]. Atomistic simulations performed for the case of PE showed that a number of different localized conformational defects cause ICD, a result consistent with the weak thickness dependence and broad distribution of $\langle \tau_c \rangle$ observed in our experimental results on PEO and POM[51].

Going beyond these existing observations we suggest that generally the large differences in $d_c$ found between different polymer systems and for different $T_c$ in the case of crystal-mobile polymers are caused by a kinetically controlled thickening process based on crystal reorganization due to ICD taking place practically simultaneously with crystal growth. In accordance with this scenario, Toda recently observed similar morphological differences between a crystal-fixed (PBT) and a crystal-mobile polymer (PE) with fast ICD as described here, actually without reference to our classification[52]. The values of the crystal thickness reported in the literature for other crystal-fixed polymers[17] are similar as observed here for PCL (PET[53]: $d_c \sim 3$ nm to 6 nm, PBT[54,55]: $d_c \sim 6$ nm to 7 nm, sPP[56]: $d_c \sim 7$ nm to 8 nm). In all these cases time-resolved SAXS shows no lamellar thickening[53,54,56]. PE, on the other hand, is a well studied crystal-mobile polymer; lamellar

thickening has been reported[57–59] and $T_c$ dependent values in the range of $d_c \sim 20$ to 60 nm[60] suggest that PE behaves similar to PEO. Various NMR studies have indeed verified a comparably fast ICD in this polymer[23,61–63]. In this scenario, crystallization of PE in the mobile hexagonal high-pressure phase that leads to crystal thicknesses on the micrometer scale would be the extreme case. Compared to the orthorhombic phase occurring under normal conditions, the ICD in this phase is about three orders of magnitude faster going along with a high conformational and partially positional disorder[64,65].

In conclusion, the experiments on poly(oxymethylene) confirm our central hypothesis that the morphology of semicrystalline polymers results from an interplay or competition of crystal growth and crystal thickening due to ICD. NMR measurements confirmed that in this polymer the ICD takes place on an intermediate timescale. The results complement our previous experiments on PCL and PEO, which correspond to the cases where the ICD is either very slow or fast in comparison to crystal growth. Across all three polymer systems, we observed a systematic increase of the lamellar thickness, which for POM and PEO follows smoothly the increasing ratio of the characteristic times $\tau_{lc}/\langle \tau_c \rangle$. These findings explain on the one hand the large differences in crystal thickness of different sample systems, on the other hand, the well-established strong dependence of $d_c$ on the crystallization temperature for many common polymers that are crystal-mobile. Thus, a unifying picture of the crystallization process emerges, in which crystallization starts with an initial thin crystal, whose thickness corresponds to marginal thermodynamic stability, and continues with a kinetically controlled lamellar thickening, resulting in a further thermodynamic stabilization. The thickening is enabled by the existence of ICD, which is a typical property of polymer crystals.

An important observation is the strong slow-down of the initially fast thickening of POM at high $T_c$ during crystallization ending up in the known very slow logarithmic growth. This fact together with the observation that reorganization always leads to a well-defined, narrowly distributed amorphous thickness, indicates that crystal thickening is increasingly restrained or suppressed by constraints in the amorphous regions, namely the entanglements. The crystal thickness observed in a fully crystallized sample is therefore strongly dependent on the ratio $\tau_{lc}/\langle \tau_c \rangle$, i.e., on the time available for more or less unhindered reorganization. Previous explanations, based on an assumed significant slowing down of the ICD due to increasing crystal thickness, are inconsistent with our experimental result that $\langle \tau_c \rangle$ is either very weakly dependent on $d_c$ (PEO) or independent of $d_c$ (POM).

Generally our results highlight the fact that the semicrystalline morphology is a non-equilibrium structure and to a large extent controlled by reorganization. Specifically, for crystal-mobile polymers, it will in general not be possible to separate crystal growth and crystal reorganization, which makes the observation of the initial crystal thickness difficult if not impossible. Theoretical predictions or interpretations of the crystal thickness, which do not take into account crystal thickening as, e.g., in the kinetic models by Hoffman–Lauritzen or Sadler[4,6,7], can therefore not be applied to crystal-mobile polymers. The immediate reorganization has the effect that the typical morphology obtained after isothermal crystallization does not reflect the kinetic barrier that is assumed to limit the crystal thickness. As mentioned, crystal thickening has been observed before and attempts have been made to include it into the Hoffmann–Lauritzen model. However, these approaches were based on specific experimental observations and gave the impression of a certain ad hoc character, c.f., e.g.,[66,67]. We showed that it is important to not only consider the crystal thickness but also the thickness of the amorphous regions, which is the better-defined parameter for crystal-mobile polymers. The results indicate that, opposite to existing approaches[9–14], it is the competition between

crystal growth and crystal reorganization and finally, the internal structure of the amorphous regions which limits the crystal thickness and therefore the crystallinity of crystal-mobile polymers. On the other hand, crystal-fixed polymers without or with very slow ICD show a well-defined crystal thickness as a function of supercooling, but the dependence is much weaker than for crystal-mobile polymers. Whether the Hofmann-Lauritzen model can explain the crystallization kinetics for these polymers is under discussion[11]. Furthermore, the crystals in these systems display only marginal thermodynamic stability directly after crystallization, which takes away one of the main observations taken as initial evidence for a postulated intermediate mesophase, whose stability with respect to the crystal phase is assumed to determine the crystal thickness in the multistage model by Strobl[11]. However, in line with the multistage models, reorganization plays a large role in polymer crystallization, but in the systems, we investigated it takes place in the crystal phase itself and is based on ICD, similarly as suggested in recent simulations[14]. While we could already show in a previous publication that in crystal-fixed polymers the selected thickness of the amorphous regions goes along with an increase of the entanglement concentration by about a factor of two compared with the melt[41], a more detailed investigation of entanglement effects on the morphology especially for crystal-mobile polymers is still lacking. From our results, one would expect that entanglements are partially resolved during thickening by ICD, a prediction that would be interesting to demonstrate directly in the future. Such studies would also take up results from simulations[68] and analytical theoretical work[69] in which the role of entanglements is considered.

## Methods

**Materials.** As model systems we chose poly(-ε-caprolactone) (PCL), poly(oxymethylene) (POM) and poly(ethyleneoxide) (PEO). PEO with a very fast and PCL without or very slow ICD[17,34,37] was already investigated in detail in previous publications[21,22]. POM shows a comparably slow ICD[17]. The sample characteristics are given in Table 1. For each sample system, we investigated two different molecular weights in order to exclude any special molecular weight effects and to show the generality of the results. The samples were named after the molecular weight $M_w$, which was determined by GPC-analysis. For PCL a polystyrene calibration and THF as solvent was used. For PEO a polystyrene calibration and $H_2O$ with 0.5 g/l $NaN_3$ as solvent was used. The poly(oxymethylenes) are industrial samples, containing a not further specified amount of stabilizers to avoid degradation by the mechanism observed by Kern and Stohler[70,71]. The molecular weight was determined using poly(-methylmethacrylate) calibration and HFIP/ 0.05 M KTFAC as solvent. All polymers have a molecular weight $M_w$ well above the entanglement molecular weight $M_e$ and the contour length $R_{max}$ is much larger than the typical size of the semicrystalline structure. Hence, the chosen polymers are representative of crystallization from an entangled polymer melt. Melting temperatures $T_m$ and crystallinity $X_c$ given in Table 1 were determined by DSC; $X_c = \Delta H/\Delta H_{100}$. Here, $\Delta H$ is the measured melting enthalpy and $\Delta H_{100}$ the extrapolated melting enthalpy for a 100% crystalline sample (POM: $\Delta H_{100} = 326$ J g$^{-1}$ [72]; PCL: $\Delta H_{100} = 157$ J g$^{-1}$ [73]; PEO: $\Delta H_{100} = 196.6$ J g$^{-1}$ [74]). As POM is sensitive for degradation at high temperatures, the isothermal crystallization step during sample preparation was performed either under nitrogen atmosphere (NMR and DSC) or under vacuum (SAXS).

### Instruments and data analysis

*Small-angle X-ray scattering.* SAXS measurements were performed on a Kratky compact camera from AntonPaar GmbH equipped with focusing X-ray optics from AXO Dresden GmbH and with a 1D detector Mython2 R 1K from Dectris. A temperature-controlled sample holder enabled in-situ isothermal crystallization experiments at different crystallization temperatures $T_c$.

The data were analyzed using a quantitative approach based on modeling the interface distribution function. A short account of the method is given in the Supplementary Information, further details can be found in refs. [21,38]. The analysis provides the mean thicknesses of the crystalline ($d_c$) and amorphous ($d_a$) domains together with their distribution widths $\sigma_c$ and $\sigma_a$ in terms of assumed Gaussian distributions as well the Porod parameter $P$. The width of the window function used to smooth the interface distribution functions was 1.0 nm for PCL, 0.8 nm for POM, and 1.4 nm for PEO.

*¹³C CP MAS CODEX.* Rotor-synchronized CODEX (center band-only detection of exchange) experiments were performed on a 400 MHz Bruker Avance system with

a ¹³C Larmor frequency of 100.6 MHz using double and triple resonance probes at a spinning rate of 5000 ± 3 Hz. During the evolution and the acquisition of the ¹³C signal high power proton decoupling (SPINAL64) was used. The $\pi$/2-pulses of the ¹H and ¹³C were set to 3.0 and 3.3 μs, respectively. The recoupled evolution time $N\tau_R$ was set to 1.2 ms with a MAS rotor period $\tau_R = 200$ μs and $N$ as an even integer number. The recycle delay $d_1$ (time between successive scans) and cross-polarization time were 8–16 s and 900 μs, respectively.

The CODEX technique[33] probes slow reorientations of the ¹³C chemical shift anisotropy (CSA) tensor and has often been used to investigate slow dynamics in semicrystalline polymers[34]. In this experiment[33] the CSA is refocused by rotor-synchronized $\pi$-pulses during an evolution and a reconversion period which are separated by a mixing time $t_{mix}$. If no reorientation of the CSA tensor occurs during the mixing time, the signal will be completely refocused. Molecular motions during the $t_{mix}$ lead to a signal decay of the exchange intensity $S_{ex}$. A reference signal $S_0$ is acquired with a short $t_{mix}$ to compensate for signal losses caused by relaxation effects. In Fig. 6a, the effect of the mixing time on the exchange and reference signal is shown. The crystalline exchange signal is reduced by relaxation effects and signal losses caused by the reorientation of the CSA tensor, the difference between reference and exchange signal increases for longer $t_{mix}$ and higher temperatures (faster monomer jump dynamics). The area under the amorphous peak (green area in Fig. 6a) is similar for both signals $S_{ex}$ and $S_0$ and is only affected by relaxation effects. To analyze the correlation time $\langle\tau_c\rangle$ describing the intracrystalline dynamics, the signal decay $S_{ex}/S_0$ is fitted based upon

$$S_{ex}(t_{mix})/S_0(t_{mix}) = p + (1-p) \cdot \exp\left[-(t_{mix}/\tau_c)\right] \quad (4)$$

with $p = 1/M$ with $M$ distinguishable sites ($M = 9$ for the $9_5$-helix in POM) as shown in Fig. 6b. In addition, we assume a lognormal distribution of the helical jump correlation time calculated numerically during the fit. The distribution width $\sigma$ attains values between 1.3 and 2.0, corresponding to a distribution extending over 1–2

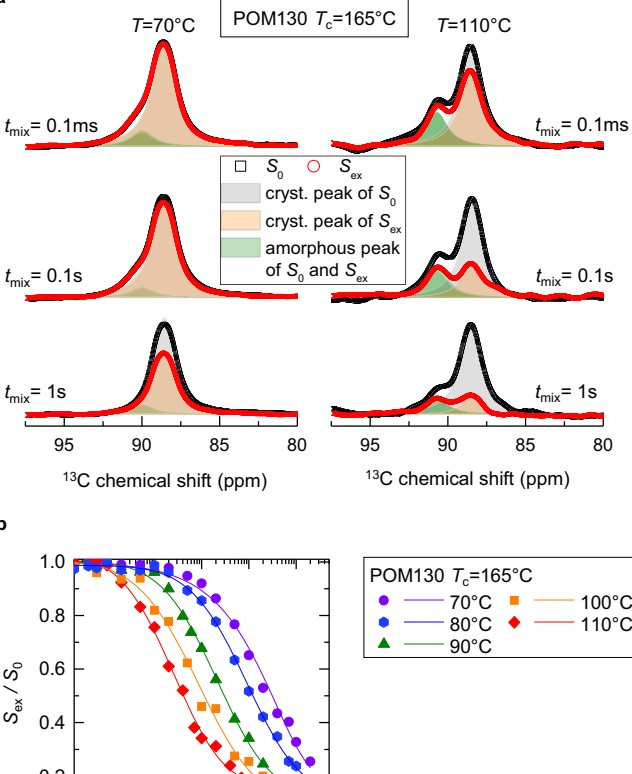

**Fig. 6 Exemplary NMR analysis on the POM samples. a** ¹³C CP MAS spectra of the exchange $S_{ex}$ (red) and reference signal $S_0$ (black) for three different mixing times at two temperatures. The resonances are deconvoluted into the amorphous (green) and crystalline exchange (orange) and reference (grey) peak. **b** CODEX $t_{mix}$ decay curves for four different temperatures resulting in the Arrhenius temperature dependence shown in Figure 1c.

decades. With regards to possible contributions from spin diffusion, which is largely temperature-independent and very slow but measurable for $^{13}$C at natural abundance, we found apparent, possibly spin-diffusion dominated values for $\tau_c$ of order 100 s and above at temperatures of 30 °C and below. These lower-limit estimates are one order of magnitude larger than $\tau_c$ in the temperature range of interest at $T > 60$ °C. Therefore, we can safely ignore spin diffusion. See also the SI (Supplementary Fig. 1).

*Polarisation microscopy.* Polarization microscopy experiments were performed on an Olympus BX51 microscope equipped with a Linkam hot stage THMS600, temperature controller TP94 and liquid nitrogen controller LNP. Samples were held between two glass slides and had a thickness of several 10 up to 80 μm. After fast cooling from the melt to different crystallization temperatures $T_c$, a series of images was recorded during isothermal crystallization and the growth velocity of spherulites was determined from their time-dependent area. For every temperature, an average of over three different spherulites was performed.

*Differential scanning calorimetry.* DSC measurements were performed with a UNIX DSC 7 from Perkin Elmer. Nitrogen was used as a purging gas, temperature calibration was performed with mercury.

## Data availability

All processed data necessary to evaluate the conclusions in the paper are provided in the paper and/or the Supplementary Information. The data shown in Fig. 3, Fig. 4a, and Fig. 5 are provided in digital format in the Supplementary Information/Source Data File. Further datasets (raw data) generated in this study are available from the corresponding authors on reasonable request. Source data are provided with this paper.

## Code availability

The Matlab Code used for the analysis of the SAXS data is available from T.T.-A. on reasonable request. Possibilities for personal instruction on using the code are limited and depend on the availability of staff.

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

## Acknowledgements

We thank Katrin Herfurt for technical help with calorimetric measurements and Anika Wurl for performing the rheological measurements. Funding was provided by Deutsche Forschungsgemeinschaft (DFG, German Research Foundation)-Projekt-ID 189853844-TRR 102, project A1 (K.S. and T.T.-A.).

## Author contributions

M. Schulz performed and designed experiments (SAXS and optical microscope), analyzed and visualized data. M. Schäfer performed and analysed the NMR experiments. T. Thurn-Albrecht and K. Saalwächter conceived and supervised the project. M. Schulz and T. Thurn-Albrecht wrote the paper. All authors discussed results and commented on the paper.

## Funding

## Competing interests

The authors declare no competing interests

## Additional information



