## [Peer Review File · Nature Communications]

REVIEWER COMMENTS

Reviewer #1 (Remarks to the Author):

This manuscript contributes convincing answers to long-standing fundamental questions about the structure of semicrystalline polymers: Why do polymers usually not crystallize as fibrous extended-chain crystals but instead form thin crystallites and amorphous layers roughly perpendicular to the chain direction? Why does the dependence of that thickness on crystallization temperature differ significantly for different polymers? Under what conditions CAN certain polymers form extended-chain crystals? This work also highlights that the community over many decades has failed to distinguish between the crystallization behavior and crystallized structure of crystal-fixed and crystal-mobile polymers. While the majority of semicrystalline polymers are crystal-fixed, high-density polyethylene and isotactic polypropylene are not, and they have outsized importance both in applications and fundamental studies. This means that a large body of crystallization studies that incorrectly extrapolated their findings to polymers in general will need to be re-assessed after the present work has been published.

The important conclusions made here are the culmination of years of work by these authors (see, for instance, p. 11, line 199: “The observations for PCL and PEO confirm and extend our previous results.” And line 201: “The results for POM now clearly establish the timescale of ICD ... as the relevant effect.”, confirming the previous hypothesis). The authors do acknowledge this related previous work already in the Abstract. This raises some issues of novelty, since significant parts of the story were published before. This is mostly a policy and not a scientific question, so I leave it to the editor to decide. The authors highlight (as mentioned twice in the Abstract) that Figure 4B contains important new data from POM. I do expect that this ‘capstone’ paper will be highly cited, and in this way publication in Nat. Commun. appears fully justifiable.

I would urge the authors to replace “intracrystalline chain dynamics” with “intracrystalline chain diffusion”. For instance, there is intracrystalline chain dynamics in nylons, due to librations of the methylene groups (Hirschinger et al., 1990, <https://doi.org/10.1021/ma00210a009>), but it is irrelevant here. It is specifically the chain diffusion dynamics that are crucial.

The authors should also reconsider the title. It is too long yet misses important aspects. The paper is really focused on crystal-mobile polymers. I would suggest: “Lamellar thickness in crystal-mobile polymers: Competition between crystal growth and chain diffusion”. If crystal-fixed polymers are meant to be included, the intriguing final discussion on this topic should be expanded and reflected in the title: “Lamellar thickness in semicrystalline polymers: Crystal growth, chain diffusion, and entanglements”.

I think it would be good to add a sentence pointing out that this work does not relate to lamellar thickness controlled by the length of the typical crystallizable chain segment, as would be the case in branched polyethylenes or vinyl polymers of one preferred, but not perfect, tacticity.

Crystal thickening requires not just diffusion of chain segments through the crystalline lamella but specifically the diffusion of chain ends. Therefore, I would expect a fairly strong effect of molecular weight here. Has this been considered? One could argue that d_c in eq.(3) should be replaced with a certain fraction (maybe 1/3?) of the contour length of the chain.

The classical crystallization theory of polymers is Hoffman-Lauritzen theory. It is surprising that it is not prominently mentioned and discussed in this work. Maybe the authors are highly critical of H-L theory, but given its prominence, it would be good for the community if the implications of the findings here for H-L theory were discussed. In line 325, opposing “existing views” are mentioned but none are quoted – this should be changed. Similarly after “postulated intermediate mesophase” in line 331. Somewhat related, in line 323 the phrase “as has often been done in the past.” begs for several prominent references (even though increasing a paper’s citation count for being incorrect may feel wrong). In line 333, after “multistage models”, some should be quoted.

line 334: Why do the authors refer to their materials as “model” system? Either it should be explained what is simplistic about these systems, or the word should be deleted.

The English in the manuscript is imperfect in many little ways that do distract from the substance of the science discussed. For instance, the text contains quite a few “German commas”. An (incomplete) list of other problems is provided here:

Line 2/3: “varies largely” should be changed to “varies significantly”.

Line 43 should say “relates the crystallinity of a polymer to the existence”

Line 50 should say “of which one is crystal-fixed while the other one is crystal-mobile”

Figure 1: Please explain red vs. black in the figure.

Caption of Figure 3, last line: “Structure parameters” should be “Structural parameters”?

Caption of Figure 3, last line: “of the y-axis” the left one only.

Line 67: “large temperature range” should be “wide temperature range”.

“previous our studies”

“a intermediate”

“time range” should be “time scale”

“in case of PCL” needs to be changed to “in the case of PCL”. This error occurs multiple times.

Line 111: “Additional investigations”: Kentgens et al. should be given credit for the first elucidation of the ICD in POM. Schmidt-Rohr and Spiess reported particularly pretty spectra, but that was later. The comonomer content in Hostaform is so low that it is not important here. For instance, the comonomer produces quite negligible NMR signal.

Line 146: “is lying” needs to be changed to “lies”.

Line 168: “in Figure 2 for PCL and POM the samples are from the same batch ...” should be changed to “in Figure 2. The PCL and POM samples are from the same batches ...”.

Line 200: Add “, respectively,” after “fast ICD

Line 284: “thickening is reported” should be changed to “thickening has been reported”.

Line 285: “Different NMR studies” should be changed to “Various NMR studies”.

Line 287: “the hexagonal phase” Clearer: “the mobile hexagonal phase”.

Line 297: “for a first time” should be changed to “for the first time”.

Line 337: Please move “crystal-fixed” to the preceding line: “that in crystal fixed polymers the selected thickness...”

Line 338: “by about a factor of two,” relative to what? The melt? Please specify.

Line 340: “entanglements are partially dissolved” – wouldn’t “resolved” or “removed” be better?

Line 349: “comparably slow” should probably be changed to “relatively slow”.

Reviewer #2 (Remarks to the Author):

The manuscript by Schultz et al. reports an interesting investigation on the origin of lamellar thickness selection in semicrystalline polymers. In particular, using POM, a polymer which displays relatively slow intra-crystalline chain dynamics (ICD), they could fill the gap with previous literature and highlight the role of the competition between crystal growth and lamellar thickening governed by ICD. The time scales of the two processes were determined by optical microscopy and NMR

methods, respectively, while the nano-scale semicrystalline morphology was investigated by time-resolved SAXS. A comparison between the new results obtained with POM with the previous literature on PCL (very slow ICD) and PEO (very fast ICD) showed that the influence of helical jumps occurring in the crystals during crystallization increases in the order $PCL < POM < PEO$. For POM, the time to grow a crystalline monolayer at low crystallization temperatures is comparable to that required for propagating a conformational defect along the lamella, while it is longer at high temperatures. As a result, the lamellar thickness of POM shows remarkable variation with crystallization time and at different crystallization temperatures. The extent of variation is in between that of PCL (negligible variation, slow ICD) and PEO (largest variation, very fast ICD).

The most notable result is the clear dependence of the lamellar thickness on the ratio of the time scales for crystal growth and ICD for crystal-mobile polymers, while crystal-fixed polymers are off the individualized trend.

The work presents noteworthy results, of significance for the community of polymer crystallization and polymer physics in general. The topic of lamellar thickness selection in polymer crystallization is in fact still debated, despite several decades of studies. The originality lies in using a polymer with ICD time scale comparable to crystal growth, which enable going beyond the results of existing literature and drawing more general and definite conclusions. This work thus leads to a deeper understanding of the phenomenon and it is expected to trigger further investigation on the topic.

The methods used are well established (also thanks to previous contributions from the authors) and suitable to the study, as such the conclusions are adequately supported by the gathered experimental data. The manuscript is clear and well-written and it contains sufficient information for reproducing the results.

Some minor comments are provided in the following:

- 1) Equation 1, the value of 5 Ang is used for the thickness of the monolayer. Perhaps the exact value of the monolayer is known from POM crystallographic and growth rate details, although it is certainly of the order of magnitude of the used value.
- 2) The time for the helical jump depends on lamellar thickness for PEO, but not for POM. Could the authors comment on this difference?
- 3) As a possible improvement, the values of lamellar thickness versus crystallization temperature for the 3 polymers could be provided in the form of a plot similar to Figure 2 or 3 (multiple plots) in the SI, for the sake of clarity, since this dependence is discussed in the text (page 11)
- 4) In Figure 4B the normalized lamellar thickness is plotted against the ratio of time for layer growth to ICD time scale. I was wondering whether a similar trend would be obtained considering simply the rate of reorganization, i.e., the reciprocal of ICD time. In other words I believe it should be explained in a bit more details why it is really necessary to consider a competition between the two processes. Is it because the largest thickening through ICD occurs during primary crystallization, mainly? I suggest adding few sentences to better explain the concept of competition, since it is the key result.
- 5) The authors demonstrated that ICD plays a major role in crystal-mobile polymers in determining the (final) lamellar thickness, including its crystallization temperature dependence. This raises the question on the role of secondary nucleation in the thickness selection for this polymer category. It is known that Hoffmann-Lauritzen theory is limited to the initial lamellar thickness and the existence of isothermal thickening is recognized. The authors might want to add a comment on this aspect, in light of their data.

Reviewer #3 (Remarks to the Author):

The manuscript successfully correlates the lamellar thickness in three different polymers to the competition between crystal growth and intracrystalline chain dynamics. It is an important aspect to answer the question why the different polymers possess different lamellar thickness at the same undercooling. Though the previous reports have attributed lamellar thickening during isothermal crystallization process to the α -c relaxation, only the qualitative relationship is established. In this work, and the critical role of the intrachain dynamics is revealed and a quantitative analysis is proposed, which is a remarkable progress.

The three types of polymers with different time scales of intrachain dynamics were chosen to observe the change of lamellar thickness and amorphous layer thickness during isothermal crystallization. It is clearly revealed that lamellar thickening happens when the time scale of crystal growth is larger than the time scale of defect jump or when the two is comparable.

The manuscript is well organized and the conclusions are supported by the experimental results. It answers the long existing question what determines the thickness of polymer lamellar crystals. Consequently, the elegant work should be accepted.

There is still one question remaining unclear. In Eq. 3, the relaxation time of a crystalline stem is proposed to be proportional to the square of the crystalline lamellar thickness. The assumption is that each defect can move independently from other parts, which is questionable. If lamellar thickening during primary crystallization is only determined by the jump of the defect in the crystal, the lamellar thickening should be proportional to the square root of time rather than the observed trend of logarithmic of time. The latter indicates that the lamellar thickening happens with increasing activation energy with prolonged time.

Some minor points:

Line 149 on page 8, $\tau(\text{stem})$ should not appear after "respectively". Please check it.

In Fig. 4B, why not use the ratio of crystal growth time to the relaxation time of a whole stem as the horizontal coordinate? Both of them will vary with the lamellar thickness so that they can be compared for a stem.

In Table 1 in SI, the prefactor τ_0 is not a constant, making determination of the activation energy questionable.

Point-by-point response to the Reviewer's comments

We thank all reviewers for their efforts and for their positive assessment of our work.

Reviewer #1

This manuscript contributes convincing answers to long-standing fundamental questions about the structure of semicrystalline polymers: Why do polymers usually not crystallize as fibrous extended chain crystals but instead form thin crystallites and amorphous layers roughly perpendicular to the chain direction? Why does the dependence of that thickness on crystallization temperature differ significantly for different polymers? Under what conditions CAN certain polymers form extended chain crystals? This work also highlights that the community over many decades has failed to distinguish between the crystallization behavior and crystallized structure of crystal-fixed and crystal-mobile polymers. While the majority of semicrystalline polymers are crystal-fixed, high density polyethylene and isotactic polypropylene are not, and they have outsized importance both in applications and fundamental studies. This means that a large of body of crystallization studies that incorrectly extrapolated their findings to polymers in general will need to be re-assessed after the present work has been published.

The important conclusions made here are the culmination of years of work by these authors (see, for instance, p. 11, line 199: "The observations for PCL and PEO confirm and extend our previous results." And line 201: "The results for POM now clearly establish the timescale of ICD ... as the relevant effect.", confirming the previous hypothesis). The authors do acknowledge this related previous work already in the Abstract. This raises some issues of novelty, since significant parts of the story were published before. This is mostly a policy and not a scientific question, so I leave it to the editor to decide.

The authors highlight (as mentioned twice in the Abstract) that Figure 4B contains important new data from POM. I do expect that this 'capstone' paper will be highly cited, and in this way publication in Nat. Commun. appears fully justifiable.

We thank the reviewer for her/his positive remarks. Replies to her/his detailed comments are given below.

I would urge the authors to replace "intracrystalline chain dynamics" with "intracrystalline chain diffusion". For instance, there is intracrystalline chain dynamics in nylons, due to librations of the methylene groups (Hirschinger et al., 1990, <https://doi.org/10.1021/ma00210a009>), but it is irrelevant here. It is specifically the chain diffusion dynamics that are crucial.

We followed the recommendation of the reviewer and introduced the necessary changes in the revised manuscript.

The authors should also reconsider the title. It is too long yet misses important aspects. The paper is really focused on crystal-mobile polymers. I would suggest: "Lamellar thickness in crystal-mobile polymers: Competition between crystal growth and chain diffusion". If crystal-fixed polymers are meant to be included, the intriguing final discussion on this topic should be expanded and reflected in the title: "Lamellar thickness in semicrystalline polymers: Crystal growth, chain diffusion, and entanglements".

We followed the suggestion of the reviewer with a slight modification. The new title reads "Lamellar thickness in semicrystalline polymers: Competition between crystal growth and intracrystalline chain diffusion" We think that 'semicrystalline' should be appear in the title. Although entanglements play an important role in the discussion, the experimental focus lies the relation between morphology and chain dynamics.

As consequence of the change in the title, we further adapted our terminology describing the intracrystalline chain dynamics, now chain diffusion in order to give a consistent picture. This led to small changes in Fig. 1a and the corresponding parts in the text.

I think it would good to add a sentence pointing out that this work does not relate to lamellar thickness controlled by the length of the typical crystallizable chain segment, as would be the case in branched polyethylenes or vinyl polymers of one preferred, but not perfect, tacticity.

In the revised manuscript we specified that we deal with chemically uniform linear polymers.

Crystal thickening requires not just diffusion of chain segments through the crystalline lamella but specifically the diffusion of chain ends. Therefore, I would expect a fairly strong effect of molecular weight here. Has this been considered?

One could argue that d_c in eq.(3) should be replaced with a certain fraction (maybe 1/3?) of the contour length of the chain.

We agree with the reviewer that molecular weight effects could be expected. For this reason, we included two molecular weights of each polymer in our study from the very beginning. As the data show there are effects on d_c and d_a , however they are not strong. We therefore decided not to focus on these effects in the current manuscript. Given this lack of evidence, to replace d_c in eq. (3) by a fraction of the contour length would be rather speculative in our view. We therefore decided to stick to the current definition of τ_{stem} . We are currently working on experiments to study the molecular weight effects on disentanglement by intracrystalline chain diffusion in more detail and hope to present more precise evidence in the future.

The classical crystallization theory of polymers is Hoffman-Lauritzen theory. It is surprising that it is not prominently mentioned and discussed in this work. Maybe the authors are highly critical of H-L theory, but given its prominence, it would be good for the community if the implications of the findings here for H-L theory were discussed. In line 325, opposing “existing views” are mentioned but none are quoted – this should be changed. Similarly after “postulated intermediate mesophase” in line 331. Somewhat related, in line 323 the phrase “as has often been done in the past.” begs for several prominent references (even though increasing a paper’s citation count for being incorrect may feel wrong).

In line 333, after “multistage models”, some should be quoted.

In the previous version of the manuscript the concluding discussion in the last paragraph of the Discussion section did in fact not contain extensive references. Our arguments were implicitly meant to refer to the references in the introduction. Since this was obviously difficult to recognize for the reader, we revised the corresponding paragraph and added the references explicitly. The implications of our results for the different crystallization models (Hoffmann-Lauritzen, Sadler- Gilmer, Strobl) are now explicitly mentioned and discussed.

line 334: Why do the authors refer to their materials as “model” system? Either it should be explained what is simplistic about these systems, or the word should be deleted.

We deleted ‘model’ as suggested.

The English in the manuscript is imperfect in many little ways that do distract from the substance of the science discussed. For instance, the text contains quite a few “German commas”.

We checked the text again and made an effort to remove the ‘German commas’.

An (incomplete) list of other problems is provided here:

Line 2/3: “varies largely” should be changed to “varies significantly”.

Done.

Line 43 should say “relates the crystallinity of a polymer to the existence”

Done

Line 50 should say “of which one is crystal-fixed while the other one is crystal-mobile”

Done.

Figure 1: Please explain red vs. black in the figure.

In Figure 1b the black and red data points are different molecular weights of POM. This information is now added in the caption. In Figure 1c data points from a third molecular weight of POM were accidentally shown as black data points. We removed these data points.

Caption of Figure 3, last line: "Structure parameters" should be "Structural parameters"?

Done.

Caption of Figure 3, last line: "of the y-axis" the left one only.

Done.

Line 67: "large temperature range" should be "wide temperature range".

Done.

"previous our studies"

Done.

"a intermediate"

Done

"time range" should be "time scale"

Done.

"in case of PCL" needs to be changed to "in the case of PCL". This error occurs multiple times.

Done.

Line 111: "Additional investigations": Kentgens et al. should be given credit for the first elucidation of the ICD in POM. Schmidt-Rohr and Spiess reported particularly pretty spectra, but that was later. The comonomer content in Hostaform is so low that it is not important here. For instance, the comonomer produces quite negligible NMR signal.

The corresponding paragraph has been reformulated accordingly.

Line 146: "is lying" needs to be changed to "lies".

Done.

Line 168: "in Figure 2 for PCL and POM the samples are from the same batch ..." should be changed to "in Figure 2. The PCL and POM samples are from the same batches ...".

Done.

Line 200: Add ", respectively," after "fast ICD"

Done.

Line 284: "thickening is reported" should be changed to "thickening has been reported".

Done.

Line 285: "Different NMR studies" should be changed to "Various NMR studies".

Done.

Line 287: "the hexagonal phase" Clearer: "the mobile hexagonal phase".

Done.

Line 297: "for a first time" should be changed to "for the first time".

The phrase has been deleted, as it should be avoided due editorial guidelines.

Line 337: Please move "crystal-fixed" to the preceding line: "that in crystal fixed polymers the selected thickness..."

Done.

Line 338: "by about a factor of two," relative to what? The melt? Please specify.

The missing information has been added (the melt).

Line 340: "entanglements are partially dissolved" – wouldn't "resolved" or "removed" be better?

We replaced 'dissolved' by 'resolved'.

Line 349: "comparably slow" should probably be changed to "relatively slow".

Done.

Reviewer #2:

The manuscript by Schultz et al. reports an interesting investigation on the origin of lamellar thickness selection in semicrystalline polymers. In particular, using POM, a polymer which displays relatively slow intra-crystalline chain dynamics (ICD), they could fill the gap with previous literature and highlight the role of the competition between crystal growth and lamellar thickening governed by ICD. The time scales of the two processes were determined by optical microscopy and NMR methods, respectively, while the nano-scale semicrystalline morphology was investigated by time resolved SAXS.

A comparison between the new results obtained with POM with the previous literature on PCL (very slow ICD) and PEO (very fast ICD) showed that the influence of helical jumps occurring in the crystals during crystallization increases in the order PCL<POM<PEO. For POM, the time to grow a crystalline monolayer at low crystallization temperatures is comparable, to that required for propagating a conformational defect along the lamella, while it is longer at high temperatures. As a result, the lamellar thickness of POM shows remarkable variation with crystallization time and at different crystallization temperatures. The extent of variation is in between that of PCL (negligible variation, slow ICD) and PEO (largest variation, very fast ICD). The most notable result is the clear dependence of the lamellar thickness on the ratio of the time scales for crystal growth and ICD for crystal-mobile polymers, while crystal-fixed polymers are off the individualized trend.

The work presents noteworthy results, of significance for the community of polymer crystallization and polymer physics in general. The topic of lamellar thickness selection in polymer crystallization is in fact still debated, despite several decades of studies. The originality lies in using a polymer with ICD time scale comparable to crystal growth, which enable going beyond the results of existing literature and drawing more general and definite conclusions. This work thus leads to a deeper understanding of the phenomenon and it is expected to trigger further investigation on the topic.

The methods used are well established (also thanks to previous contributions from the authors) and suitable to the study, as such the conclusions are adequately supported by the gathered experimental data. The manuscript is clear and well-written and it contains sufficient information for reproducing the results.

We thank the reviewer for her/his positive remarks. Replies to her/his detailed comments are given below.

Some minor comments are provided in the following:

1.) Equation 1, the value of 5 Å is used for the thickness of the monolayer. Perhaps the exact value of the monolayer is known from POM crystallographic and growth rate details, although it is certainly of the order of magnitude of the used value.

The thickness of one molecular layer in the crystal is used to estimate the layer crystallization time τ_{lc} on a logarithmic scale, cf. Fig. 2. In this context, it makes no difference if the molecular layer has a thickness of 2, 3 or 5 Å. On the other hand, to get the exact value, the crystal lattice would be needed as well as the dominant growth direction for crystallization from the melt. As these quantities are not always known or at least often difficult to find, but typically of same order of magnitude, we decided to generally use a typical value of 5 Å instead. This choice has no effect for our general arguments.

2.) The time for the helical jump depends on lamellar thickness for PEO, but not for POM. Could the authors comment on this difference?

There is a possibly important difference between PEO and POM. In PEO we could cover a wide range of d_c values. This fact made it possible to conclude on a relatively weak dependence of $\langle\tau_c\rangle$ on d_c , although experimental accuracy is limited (cf. Ref. 33). In the case of POM, the accessible range of d_c is smaller and experimental accuracy is similarly limited. For this reason, we were careful to formulate our result, ‘we could not observe a systematic dependence of $\langle\tau_c\rangle$ on d_c .

3.) As a possible improvement, the values of lamellar thickness versus crystallization temperature for the 3 polymers could be provided in the form of a plot similar to Figure 2 or 3 (multiple plots) in the SI, for the sake of clarity, since this dependence is discussed in the text (page 11)

We followed the suggestion of the reviewer and included a new plot in the Supplementary Information (Figure 4). To save space we took the apparent supercooling with reference to the DSC melting temperature as a common x-axis.

4.) In Figure 4B the normalized lamellar thickness is plotted against the ratio of time for layer growth to ICD time scale. I was wondering whether a similar trend would be obtained considering simply the rate of reorganization, i.e., the reciprocal of ICD time. In other words I believe it should be explained in a bit more details why it is really necessary to consider a competition between the two processes. Is it because the largest thickening through ICD occurs during primary crystallization, mainly? I suggest adding few sentences to better explain the concept of competition, since it is the key result.

The smooth dependence observed in Figure 4b for the normalized crystal thickness is only observed for the combined variable $\tau_{lc}/\langle\tau_c\rangle$, not for $1/\langle\tau_c\rangle$. As the comparison is rather illustrative we included for comparison the suggested plot in the Supplementary Information (Figure 5). A corresponding remark was added on page 13. It is necessary to consider a competition between reorganization and growth, if both processes take place on similar timescales leading to reorganization during primary crystallization, as mentioned by the reviewer. We added a corresponding remark on page 11, at the point where Figure 4b is introduced.

5) The authors demonstrated that ICD plays a major role in crystal-mobile polymers in determining the (final) lamellar thickness, including its crystallization temperature dependence. This raises the question on the role of secondary nucleation in the thickness selection for this polymer category. It is known that Hoffmann-Lauritzen theory is limited to the initial lamellar thickness and the existence of isothermal thickening is recognized. The authors might want to add a comment on this aspect, in light of their data.

We already introduced a more detailed discussion of the Hoffman-Lauritzen theory in light of our results on page 16 following the suggestions by reviewer #1.

Reviewer #3:

The manuscript successfully correlates the lamellar thickness in three different polymers to the competition between crystal growth and intracrystalline chain dynamics. It is an important aspect to answer the question why the different polymers possess different lamellar thickness at the same undercooling. Though the previous reports have attributed lamellar thickening during isothermal crystallization process to the α -c relaxation, only the qualitative relationship is established. In this work, and the critical role of the intrachain dynamics is revealed and a quantitative analysis is proposed, which is a remarkable progress. The three types of polymers with different time scales of intrachain dynamics were chosen to observe the change of lamellar thickness and amorphous layer thickness during isothermal crystallization. It is clearly revealed that lamellar thickening happens when the time scale of crystal growth is larger than the time scale of defect jump or when the two is comparable.

The manuscript is well organized and the conclusions are supported by the experimental results. It answers the long existing question what determines the thickness of polymer lamellar crystals. Consequently, the elegant work should be accepted.

We thank the reviewer for her/his positive remarks. Replies to her/his comments are given below.

There is still one question remaining unclear. In Eq. 3, the relaxation time of a crystalline stem is proposed to be proportional to the square of the crystalline lamellar thickness. The assumption is that each defect can move independently from other parts, which is questionable. If lamellar thickening during primary crystallization is only determined by the jump of the defect in the crystal, the lamellar thickening should be proportional to the square root of time rather than the observed trend of logarithmic of time. The latter indicates that the lamellar thickening happens with increasing activation energy with prolonged time.

We agree that the assumption that each defect can move independently can be questioned. However, our aim is not to predict the shape of the curve $d_c(t)$, but rather to give an estimate for the time range, within which we can expect that reorganization takes place. Therefore, we explicitly mentioned that eq. (3) is an approximation for the early stage of growth. Furthermore, we point out, that the logarithmic thickening is observed on a larger timescale beyond primary crystallization. As visible in Figure 4a, there are deviations at shorter times. On the other hand, the data are not sufficient to determine the time dependence in this range. In conclusion, we think that our estimate of τ_{stem} is a simple and reasonable estimate, which serves the necessary purpose.

Some minor points:

Line 149 on page 8, τ_{stem} should not appear after “respectively”. Please check it.

Corrected.

In Fig. 4B, why not use the ratio of crystal growth time to the relaxation time of a whole stem as the horizontal coordinate? Both of them will vary with the lamellar thickness so that they can be compared for a stem.

We considered this suggestion. However in our view, our choice is more simple, and there are several reasons, why we decided to stick to our original choice. $\langle\tau_c\rangle$ is a directly measured quantity, while τ_{stem} is an approximate estimate based on certain assumptions as discussed above. Additionally, τ_{stem} has an increased uncertainty, as it combines the results of two different experimental methods (SAXS & NMR), whereas $\langle\tau_c\rangle$ can be obtained by NMR alone. Furthermore, τ_{stem} depends itself on d_c . If we analyzed the dependence of d_c on τ_{stem} , the variables would not be separated. As a last point, the lamellar thickness d_c is a time dependent parameter.

In fact, the suggested alternative variable for the x-axis would not lead to a different conclusion, as the smooth variation of the normalized crystal thickness is still observed, as

illustrated in the plot below. This result is not surprising; as Figure 2 shows the difference between τ_{stem} and $\langle\tau_c\rangle$ is approximately a constant factor.

In Table 1 in SI, the prefactor τ_0 is not a constant, making determination of the activation energy questionable.

We do not agree to this statement. As shown in Figure 1c, we fit the measured values of τ_c in an Arrhenius plot by a straight line. This gives the activation energy as the slope and $\log \tau_0$ as the intercept, both with their respective errors. Within the temperature range accessible for measurements ($1000/T$ between 2,5 K^{-1} and 3,0 K^{-1}) the fit curves describe the data well within the experimental accuracy. The error in $\log \tau_0$ corresponds to the uncertainty of an extrapolation to infinite temperature, which is of course relatively large. However, we use these results to estimate $\langle\tau_c\rangle$ in the temperature range of crystallization, which is not very far outside of the measured temperature range and therefore associated with a much smaller error. To avoid misunderstandings, we modified Table 1 in the Supplementary Information. We now present the values for $\log(\tau_0/1s)$ together with the corresponding error intervals.

Remark: In addition, we moved the text that was originally in the appendix to the last paragraph of the section 'Results', just before 'Discussion' according to the journal guidelines.

REVIEWERS' COMMENTS

Reviewer #1 (Remarks to the Author):

Review with formatting (bold and strike-out) attached.

Reviewer #2 (Remarks to the Author):

The authors have satisfactorily replied to previous comments. The manuscript is acceptable in the current form.

Reviewer #3 (Remarks to the Author):

The reviewers' comments have been fully addressed in the revision. So I suggest acceptance of the revision.

Point-by-point response to the Reviewer's comments

Reviewer #1

We thank the reviewer for her/his efforts. Replies to her/his detailed comments are given below.

The authors have made meaningful revisions in response to the reviewers' comments. The text, including the discussion, now reads quite well. However, in its details, the language still has many flaws; some are listed below. All authors, including the professors, need to invest time and read the whole text line by line in order to eliminate obvious errors like "the the", "a the", or "of the of the" as well as more subtle ones that only experienced experts will notice.

We thoroughly checked the text and corrected all language errors we could find.

One other point is the terminology for the chain motions in the crystallites. While in the manuscript title, Abstract, and Introduction, it is called "intracrystalline chain diffusion" and abbreviated as "ICD", in the captions of Figures 2, 3, and 4 it is called "alpha_c-relaxation". The term "crystal-mobile" is also used elsewhere. To the experts in this field, it's clear that all of these refer to the same process, but to outsiders this might not be obvious. The authors should consider this aspect.

In order to be more precise, we rephrased and extended the discussion of the intracrystalline chain diffusion on the upper part of page 4, where the different time scales and the terminology are introduced. For the rest of the text we rendered the terminology as uniform as possible.

Overall, as before and in agreement with the other reviewers, I find that this manuscript contributes convincing answers to long-standing fundamental questions about the structure of semicrystalline polymers. It deserves to be published in Nature Communications after the language issues have been addressed.

We appreciate the positive evaluation.

Language issues:

For brevity, we did not copy the following list of language errors. We corrected all these errors as suggested. We thank the reviewer for taking the trouble to point out the language issues.

Reviewer #2:

The authors have satisfactorily replied to previous comments. The manuscript is acceptable in the current form.

Reviewer #3:

The reviewers' comments have been fully addressed in the revision. So I suggest acceptance of the revision.

We thank reviewer #2 and #3 for their efforts and for their positive assessment of our work.